# Cure of Alzheimer’s Dementia Requires Addressing All of the Affected Brain Cell Types

**DOI:** 10.3390/jcm12052049

**Published:** 2023-03-04

**Authors:** Jeffrey Fessel

**Affiliations:** Department of Medicine, University of California, San Francisco, CA 94143, USA; jeffreyfessel@gmail.com; Tel.: +1-415-563-0818

**Keywords:** Alzheimer’s dementia, curative treatment, address brain cell types, two drugs—pioglitazone with fluoxetine, two drugs—pioglitazone with lithium, three drugs—pioglitazone/fluoxetine/clemastine, three drugs—pioglitazone/fluoxetine/memantine, three drugs—pioglitazone/fluoxetine/fingolimod, three drugs—pioglitazone/lithium/clemastine, three drugs—pioglitazone/lithium/memantine, three drugs—pioglitazone/lithium/fingolimod

## Abstract

Multiple genetic, metabolic, and environmental abnormalities are known to contribute to the pathogenesis of Alzheimer’s dementia (AD). If all of those abnormalities were addressed it should be possible to reverse the dementia; however, that would require a suffocating volume of drugs. Nevertheless, the problem may be simplified by using available data to address, instead, the brain cells whose functions become changed as a result of the abnormalities, because at least eleven drugs are available from which to formulate a rational therapy to correct those changes. The affected brain cell types are astrocytes, oligodendrocytes, neurons, endothelial cells/pericytes, and microglia. The available drugs include clemastine, dantrolene, erythropoietin, fingolimod, fluoxetine, lithium, memantine, minocycline, pioglitazone, piracetam, and riluzole. This article describes the ways by which the individual cell types contribute to AD’s pathogenesis and how each of the drugs corrects the changes in the cell types. All five of the cell types may be involved in the pathogenesis of AD; of the 11 drugs, fingolimod, fluoxetine, lithium, memantine, and pioglitazone, each address all five of the cell types. Fingolimod only slightly addresses endothelial cells, and memantine is the weakest of the remaining four. Low doses of either two or three drugs are suggested in order to minimize the likelihood of toxicity and drug–drug interactions (including drugs used for co-morbidities). Suggested two-drug combinations are pioglitazone plus lithium and pioglitazone plus fluoxetine; a three-drug combination could add either clemastine or memantine. Clinical trials are required to validate that the suggest combinations may reverse AD.

## 1. Introduction

The last five decades have seen a deluge of data concerning the genetic, metabolic, and environmental factors that contribute to the pathogenesis of Alzheimer’s dementia (AD). Birle et al. pointed out that cognitive dysfunction represents a consequence of a global imbalance within three levels of brain organization: the cellular and molecular level, circuitry level, and large-scale network level [1]. The central nervous system is bidirectionally related with other fundamental systems such as the immune, endocrine, and autonomic systems and the microbiota, all of which have been shown to change adversely in patients with cognitive dysfunctions. If all of the known abnormalities were addressed it should be possible to cure the dementia, which should be the goal of treatment. However, that would require a suffocating number of drugs. Nevertheless, the problem may be simplified by addressing, instead, the brain cell types whose functions become changed as a result of all the above abnormalities because it is these cells, whose number and functions are changed, that are ultimately responsible for AD pathogenesis. This approach to formulating pharmacotherapies for psychiatric conditions in general, was described in an earlier article [2]; here, it is described in detail for AD in particular. Fortunately, at least eleven drugs are available from which to formulate a rational therapy to correct those changes. After a very brief summary of the multiple metabolic and genetic factors that participate in the pathogenesis of AD, this article will describe the brain cell types that are affected in AD, and the available drugs that can correct those changes.

## 2. Altered Metabolism and Genetics in AD

Reactive oxygen species (ROS) have a major role in the pathogenesis of AD; their formation has several causes including mitochondrial damage. ROS link to multiple down-stream damages: to mitochondria themselves, due to reduced levels of NAD/NADH; diminished glucose uptake and response rates to insulin/IGF-1 signaling (due to decreased levels of the insulin receptor and glucose transporter 1 densities); aberrant calcium regulation; disturbed lipid metabolism; impairments of autophagic pathways affecting proteostasis; and reducing water diffusion into brain cells which impacts drug delivery.

Mutations in the *APP, PS1,* and *PS2* genes are causal for familial AD (FAD). Many other genes participate in the pathogenesis of late onset AD (LOAD). The ε4 allele of the *APOE* gene is frequently present. Patients with both FAD and LOAD who are *APOε4* carriers, have abnormally low rates of glucose metabolism in the neocortex that may also be present in carriers of *APOε4* without dementia. How *APOε4* affects different brain cell types is discussed below. Brain cells from subjects with LOAD have down-regulation of many genes affecting multiple metabolic pathways: abnormalities in glucose metabolism via the pentose phosphate pathway, glycolysis, and tricarboxylic/citric acid pathways; oxidative phosphorylation; and genes responsible for subunits of cytochrome p450 that are involved in fatty acid and cholesterol oxidation. Other down-regulated genes include those responsible for metabolism of glucose [*GPD1*, *GPDHL*, and *GPDH2*, *IGF-1* and *2*, insulin receptor gene (*INSRR*), insulin receptor substrate 2 (*IRS2*), glycogen phosphorylase (*PYGL*), phosphoglucomutase (*PGM1*, *PGM2L1*), hexokinase (*HK1* and *2*), phosphofructokinase (*PFKFB3*, *PFKP*)], lactate dehydrogenase A (*LDHA*), isocitrate dehydrogenase 1 (*IDH1*), pyruvate dehydrogenase kinase (*PDK3*)] (see Ref. [3] for full descriptions). In both Parkinson’s disease and diabetes mellitus, each of which has genetic underpinnings, there are high rates of dementia. Individuals with trisomy 21 have an extra copy of the amyloid precursor protein (*APP*) gene on chromosome 21, leading to excessive cerebral amyloid. 

As is obvious from this highly abbreviated description, even were it possible to correct the innumerable metabolic and genetic abnormalities seen in AD, the task would be impossible to achieve because correction of all of them would require an intolerable number of drugs. It would be far simpler is to correct the changed functions of the affected brain cell types.

## 3. Brain Cell Types That Are Changed in AD

Changes that are in the direction of decreased functions affect astrocytes, oligodendrocytes, neurons, and endothelial cells/pericytes; changes that result in gain of function involve microglia. The following discusses the affected cell types.

*Astrocytes in AD.* Emphasizing their importance for cognition, astrocyte numbers in the dentate gyrus were more reduced in Braak stages 3–4 than in stages 0–2 [4]. Because astrocyte processes wrap around the cerebral microvasculature, the morphological modifications of astrocytes affect micro-cerebral blood flow and, therefore, the nutrients available to neurons [5,6]; in addition, the processes of astrocytes contain aquaporin-4 (AQ4), which regulates water entry into endothelial cells [7]. The consequences of the decreased numbers of astrocytes in AD include decreases in neural function and the passage of drugs into the brain.

*Oligodendrocytes in AD.* Mature oligodendrocytes myelinate the naked axons of neurons; if their numbers are decreased, e.g., from impaired maturation of oligodendrocyte precursor cells (OPCs), so that myelination becomes inadequate, then neural tracts suffer and cognition may be disturbed [6]. Exposure of oligodendrocyte cultures to 1 μM of Aβ_1-42_ induced cell death, morphological changes with shrunken cell bodies and a breakdown of their processes, and an 3.2-fold increase of lactate dehydrogenase activity that was released into the culture media by oligodendrocytes [8].

*Neurons in AD.* AD patients with severe tau pathologies had a decreased number of newly generated neurons in the dentate gyrus [4]. Leng et al. found a selectively vulnerable subpopulation of excitatory neurons in the entorhinal cortex (EC) and showed a depletion of this subpopulation during AD progression [9]. In AD, Whitehouse et al. showed a 75% reduction in cholinergic neurons in the nucleus basalis of Meynert [10]. Tse et al. analyzed MRI data from 507 subjects including healthy controls, and persons with mild cognitive impairment {MCI} or AD; they found that, with the progress of dementia, the gray matter volume of the subjects was strongly reduced and correlated with both white matter volume and MMSE scores [11].

*Endothelial cells in AD.* In MCI, there are already decreased levels of VEGF [12], and in 55 patients with established AD, there were reduced numbers of endothelial cell precursors in the peripheral blood that were associated with lower MMSE scores and higher (i.e., worse) Clinical Dementia Rating (CDR) scores [13]. Microvascular diseases affect the progression of cognitive deficits in AD [14]. Endothelial cells contain the scavenge receptor, CD36, whose interaction with Aβ activates NADPH oxidase, producing both vascular oxidative stress and neurovascular dysregulation [15]. String vessels, the remnants of destroyed microcapillaries, are seen throughout the brain of AD patients [16]. In addition, microinfarcts, with minute foci of neuronal loss as small as 50 µm, were seen in 43% of AD brains [17].

*Microglia in AD.* Several reports concerning the role of microglia in AD are conflicting, the likely reason being the need to distinguish between proinflammatory (M1) and anti-inflammatory (M2) microglia, because a skewed M1 activation over M2 has been related to disease progression in AD [18], and although this is the dominant change in AD, it is not present in all cases. Olah et al. obtained microglia from the brains of 14 participants in a study of cognitive aging and 3 subjects with intractable epilepsy, and recognized seven clusters of microglia, only one of which was reduced in patients with AD [19]. Grubman et al. found that the *APOE* risk gene is up-regulated in an AD-specific subpopulation of microglia [20].

## 4. In Summary, the Affected Brain Cell Types in AD Are Astrocytes, Oligodendrocytes, Neurons, Endothelial Cells/Pericytes, and Microglia

Affecting the functions of all of the above brain cell types is the APOE4 variant of the APOE gene product, that renders those who carry it susceptible to AD.

The APOE4 variant of the *APOE* gene product, renders its carriers susceptible to AD. Relevant to the thesis of this article, one mechanism for this may be its effect on the five affected brain cell types. Blanchard et al. reported several such effects: cholesterol transport was defective in oligodendrocytes that were derived from induced pluripotent stem cells (iPSCs) carrying APOE4; these oligodendrocytes accumulated cholesteryl ester species this was associated with down-regulation of genes linked to myelination, so that the oligodendrocytes had decreased production of Myelin Basic Protein (MBP) [21]. Besides affecting oligodendrocytes, Lin et al. showed that the *APOE4* variant can also lead to extensive gene expression alterations in neurons, astrocytes, and microglia; *APOE4* astrocytes had cholesterol accumulation, and APOE4 microglia-like cells had increased up-regulation of genes associated with immune responses and inflammation [22]. Victor et al. saw lipid accumulation in *APOE4+* microglia, and found that medium taken from cultures of *APOE4* microglia, exacerbated the pro-inflammatory activity compared with that taken from cultures of *APOE3* microglia [23]. Regarding endothelial cells, those derived from *APOE4+* pluripotent stem cells had dysfunctions in pathways leading to effects on blood clotting factors and inflammation [24].

Summary: the dominant changes in brain cell-type in AD are decreases in astrocytes, oligodendrocytes, neurons and endothelial cells, and increases in microglia; those changes may in part be caused by the presence of *APOE4*. 

## 5. Available Drugs That Address the Changes in Brain Cell Types That Underpin the Pathogenesis of AD

Fortunately, there are several drugs that correct the changes in the affected brain cell types, and from which we can formulate a rational treatment regimen. It is recommended to choose a combination of two drugs from the following list, in order to both reduce the dosages and minimize the likelihood of adverse events.

Highly abbreviated descriptions for likely therapeutic effects are provided here: the drugs either induce an increase in numbers or activation of astrocytes, oligodendrocytes, synapses and neurons, and endothelial cells; or a decrease in numbers or activation of microglia. 

**Clemastine** increased astrocytes [25], postsynaptic proteins [25], muscarinic receptor, synapsin 1 and Homer 1, and oligodendrocytes survival/function [26,27]. It also improved myelin repair [28] and neuronal function [29] and enhanced myelination in the prefrontal cortex (PFC) [30] and enhanced visual function [26,31]. Clemastine provided mitochondrial protection [32,33] and suppressed microglial M1 activation [34].

**Dantrolene**, by antagonizing ryanodine receptors and blocking Ca^2+^ release, prevented the detriment to astrocytes caused by excessive glutamate [35] It also prevented oligodendrocyte death [33,36], and preserved synaptic and neural function. References [37,38,39,40,41,42,43,44,45,46,47,48] showed several other mechanisms for neuroprotection by dantrolene. A potential adverse effect is via the cerebral microvasculature, because release of brain derived neurotrophic factor (BDNF) by endothelial cells requires Ca^2+^ mobilization [36].

**Erythropoietin** increased differentiation of neuronal stem cells (NSCs) into astrocytes; increased oligodendrocytes [49] and neurons and dendritic spines [50] via increased production of BDNF and its receptor [51]; and it also did so by enhancing differentiation of NSCs [52]. Erythropoietin improved the functioning of synapses [53,54,55], endothelial cells [56,57], and microglia [58].

Fingolimod, which is a pro-drug for an agonist of sphingomyelin phosphate (S1P), induced cell proliferation [59]. In astrocytes, fingolimod activated neurotrophic genes [60], and reduced formation of ceramides that cause apoptosis [61,62]. It activated myelinating oligodendrocyte [63], increased OPCs, myelination, and neurological function, and ameliorated brain demyelination [64]. It enriched synaptic genes [65], prevented synaptic toxicity [66], and reversed synaptic hypersensitivities [66]. It produced myelin in brains that had been demyelinated [67], and prevented neural death from N-methyl-D-aspartate (NMDA) [68]. In multiple sclerosis, it improved axonal and myelin integrity [69]; and in mice it prevented demyelination [70]. It increased dendritic spines [71,72], reduced neuronal death from reactive oxygen species (ROS) [73], and improved mitochondrial production of ATP [74]. Weak evidence shows some benefit to the cerebral microcirculation: fingolimod prevented BBB disruption induced by sera from patients with multiple sclerosis [75].

In microglia, fingolimod shifted M1 polarization toward M2 [76,77].

**Fluoxetine** induced increases in: astrocytes [78], oligodendrocytes [79,80,81], neurons [82,83,84], and endothelial cells [85], and decreased microglial activation [81,86].

Fluoxetine activated astrocytes to produce BDNF [78,87,88] and promoted clearance of astrocytes with damaged mitochondria [89]. It up-regulated oligodendrocyte precursor cells (OPCs) and oligodendrocyte markers [79], and reduced oligodendrocyte senescence [90]. Fluoxetine increased neurogenesis [79,82] and neuronal circuits [84]. Neurons deprived of glucose and oxygen had increased survival with fluoxetine [83]. It increased vasodilatation via differentiation and proliferation of endothelial cells and decreased arteriolar tone [85,91]. In microglia, it attenuated NADPH oxidase activation, production of ROS and reactive nitrogen species, and down-regulated M1 and up-regulated M2 activation [92,93].

**Lithium** doubled astrocytic numbers and their VEGF secretion [94]. It increased oligodendrocyte expression of proteolipid protein (PLP) and myelin basic protein (MBP), improving synaptic and neuronal function [95,96,97,98,99]. By negating activation of GSK-3β, synaptic expression of post synaptic density (PSD)-95 and gephyrin were enhanced [100]. Lithium promoted neurogenesis [99,101,102,103], doubled BDNF levels, increased dendritic length, increased anti-apoptotic Bcl2 and Bcl-_XL,_ and prevented neuronal death caused by excessive glutamate, pro-apoptotic BAD and BAX, and caspases [104]. It increased the number and size of neuronal mitochondria [105], increased antioxidants [97,105], minimized neurotoxicity from cytochrome c, and promoted mitochondrial biogenesis [106,107]. Lithium benefitted the microvasculature by increasing endothelial cell survival [108], enhancing VEGF secretion [109], and increasing BBB integrity [110]. Lithium’s inhibition of GSK-3β reduced production of microglial [111,112] and activation of pro-inflammatory mediators [113].

**Memantine** induced activation of astrocytes [114] and it prevented losses of oligodendrocytes [114], synapses and neurons [115,116,117], and endothelial cells [118]. It is a NMDAR antagonist, enters the NMDAR’s Ca^2+^ channel and decreases its permeability, so prevents neuronal excitotoxicity and death [115]. Memantine provides synaptic protection via several mechanisms [119,120,121], preventing cytotoxicity by blocking inhibition of a guanosine triphosphatase involved in multiple cellular processes, thus protecting against cell death caused by mitochondrial dysfunction that leads to cytochrome c release, and ROS and peroxide production [122,123,124,125]. Blocking the ion channel of the acetylcholine receptors also prevents neurotoxicity [126]. Memantine benefitted brain endothelial cells and blocked disruption of the BBB. Microglial function improves because memantine up-regulates the M2 isotype [127].

**Minocycline** prevented oligodendrocyte toxicity caused by deprivation of oxygen and glucose [128], by microglial-induced apoptosis [129]. Minocycline also prevented Aβ inhibition of cAMP-responsive element-binding protein (CREB) up-regulation [119,130]. Synaptic function improved from increased levels of PSD-95 and dendritic spines [121]. It prevented cognitive decline caused by an antagonist of the NMDAR [120]. Neuronal benefit was also derived from its increased expression of BDNF, CREB, and phospho-CREB [119], proliferation of neuronal precursor cells (NPC) [127], and potentiation of neurite outgrowth [131]. In a transgenic mouse model of Down’s syndrome, minocycline prevented the decline of cholinergic neurons [132]. In a list of 1040 drugs that prevent the release of cytochrome c (which induces a series of biochemical reactions that result in caspase activation and subsequent cell death), minocycline was the second most potent [133]; the mechanism involves inhibiting mitochondrial increases in Ca^2+^ concentration plus inhibition of NADH-cytochrome c reductase and cytochrome c oxidase [133,134]. Finally, minocycline prevented microglial activation [135,136], production of pro-inflammatory cytokines [137], and promoted M2 microglial polarization [127].

**Pioglitazone**, a peroxisome proliferator-activated receptor gamma (PPARγ) agonist, prevents phosphorylation of JAK–STAT in astrocytes and thereby induces increases in neurons [138,139], oligodendrocytes [139,140], endothelial cells [141], and decreases in microglia [142]. Other PPARγ agonists, ciglitazone and curcumin, were also cytoprotective for astrocytes and reversed their decreased expression of PPARγ receptor after exposure to Aβ_25–35_ [143,144]. Ciglitazone also increased formation of oligodendrocyte progenitors [139]. Pioglitazone rescued the demyelination caused by anti-MOG autoantibodies [145]. The promotion of OPC differentiation into mature oligodendrocytes by IL-4 was mediated by PPARγ or curcumin [140,142]. PPARγ agonism increases the neurogenic differentiation gene *NeyroD1* [139] and protected cortical neurons and axons against toxicity induced by NO or KCl [138,146].

PPARγ also induces endothelial cell proliferation and angiogenesis [141,147]. In microglia, PPARγ agonists inhibit cytokine production by down-regulating pro-inflammatory genes [148]. In addition, the PPARγ agonist rosiglitazone up-regulates M2 microglia [149].

**Piracetam** increased the numbers and functioning of astrocytes [150]. It also increased the function of synapses [151,152,153] and neurons [151]: it decreased neurotoxicity from deprivation of oxygen and glucose, hypoperfusion, ethanol feeding, or ethanol withdrawal [152,154,155,156]. It also caused longer neurites [151]. The effects of piracetam may derive in part from the restoration of cell membrane fluidity induced by a conformational change in the phospholipids of the liposomal membrane [157]. By decreasing mitochondrial swelling and permeability caused by excessive Ca^2+^ opening the mitochondrial permeability transition pore (MPTP), it improved mitochondrial membrane potential and ATP levels, shifted the balance of mitochondrial fission or fusion towards fusion, and reversed the adverse effects of pro-oxidants [158]. Additionally, it reversed the cytotoxic effects of p53 and BAX [152].

**Riluzole** increased both the gene and protein expression of the excitatory amino acid transporter (EAAT2), thus enhancing glutamate uptake by astrocytes, which protects against excitotoxicity of neurons [159,160]. It benefits synapses as shown by its inhibition of voltage-activated sodium currents which prevents reverse operation of the Na^+^/Ca^2+^ exchanger [161], and by increasing Long-Term Potentiation (LTP) [162]. It caused a 40-fold increase in hippocampal BDNF, inducing neurogenesis [163] and protected against neural degeneration caused by ischemia [164]. N-acetylaspartate is exclusively expressed in neurons and was increased in the cerebral cortex due to exposure to riluzole [165]. In neurons, it also decreased oxidative stress, lipid peroxidation, and ATP depletion [166,167]. Microglial activation was ameliorated by riluzole [168] which also upregulated the mRNA levels of M2 markers and downregulated those of M1 markers [169].

## 6. Adverse Effects

The choice of a medication is partly determined by its safety profile, particularly for serious adverse events (SAEs). The following are summaries of reports of the SAEs for the three drugs that are suggested here for use in clinical trials.

*Fluoxetine*: Beasely et al. obtained data from 25 double-blind clinical trials involving 4016 patients with MDD randomized to treatment that included fluoxetine 20 to 80 mg/d [170]. At a dose of 20 mg/d, fluoxetine-treated patients had a discontinuation rate due to adverse events that was not significantly different from that in placebo recipients. In this article, the suggested dose is 10 mg/d in combination with lithium.

*Lithium*: SAEs from lithium include hypercalcemia, hypothyroidism [171], nephrogenic diabetes insipidus, and renal insufficiency [172]. Low dosage (<100 mg daily) benefited Alzheimer’s dementia [173]; thus, 75 mg daily might be an appropriate dose in BD, MDD, SCZ, AD, or PTSD. It should not be used if the glomerular filtration rate is <60 mL/min.

*Pioglitazone*. The main adverse effects reported with pioglitazone are congestive heart failure, pedal edema and weight gain, and bone loss. Lincoff et al. performed a meta-analysis of 19 trials enrolling 16,390 patients receiving pioglitazone or comparator for 4 months to 3.5 years. Death, myocardial infarction, or stroke occurred in 4.4% of participants receiving pioglitazone and in 5.7% receiving the control therapy [174]. The number of neoplasms was similar in the pioglitazone and placebo arms of the trials. In a review, Betteridge noted sustained improvements in serum levels of triglycerides and HDL cholesterol and favorable effects on the particle size of LDL cholesterol [175]. The PROactive trial saw manageable increases in edema (26.4% vs. 15.1% for placebo) and weight gain (+3.8 kg vs. −0.6 kg for placebo); and, despite more heart failure in the pioglitazone group (5.7% vs. 4.1% for placebo), there was a proportional improvement in macrovascular outcomes among the patients who developed heart failure, and whose absolute rates of macrovascular events and mortality were similar to those in the placebo group. The incidence of malignancies were similar in the pioglitazone versus placebo arms of the study, but a higher rate of bone fractures was observed among pioglitazone-treated female patients (5.1% vs. 2.5%) [176]. This profile suggests that adverse events are outweighed by favorable outcomes.

## 7. Discussion

The cure of established AD requires addressing the cerebral cell types that are responsible for its pathogenesis. Those cell types are astrocytes, oligodendrocytes, neurons, endothelial cells/pericytes, and microglia. While it is not necessarily the case that, for individual subjects with AD, all five cell types are involved, for practical purposes it is prudent to assume that they are. Therefore, drugs that address all five cell types are the best candidates for a regimen that has the best chance for successful reversal of the dementia. It is recommended to use a combination of two drugs in order to minimize dosages and the risk of both serious adverse reactions (SAEs) and drug–drug interactions (including with drugs used for co-morbid conditions). The best chances would come from use of the drugs that address all five cell types. Those are erythropoietin, fingolimod, fluoxetine, lithium, memantine, and pioglitazone. However, unless there is compromise of the BBB, erythropoietin will not cross it; and fingolimod only weakly benefits the cerebral microcirculation. Therefore, the choice of a two-drug combination must be chosen from fluoxetine, lithium, and pioglitazone; however, fluoxetine and lithium potentially cause the most SAEs. Therefore, a clinical trial to test whether two of the suggested drugs will, in fact, cure AD should administer pioglitazone plus either fluoxetine or lithium, using low dosages: pioglitazone 15 mg/day, lithium 75 mg/day, or fluoxetine 10 mg/day. Clemastine (benefits four cell types, and causes no SAEs), fingolimod, and memantine could be added to form a three-drug combination. Clinical trials would demonstrate the validity of both the proposed concept and the choice of drugs for treatment and would also show the percentage of patients benefitting from the chosen combination of drugs. It is important to note that this use of the suggested drugs would be off-label, and that clinical trials are required to establish their efficacy for reversal of AD.

In a clinical trial that randomly assigns participants to receive active drugs or placebo, calculation of the required number of participants would be based upon the primary objective showing ≥30% more cure of AD in those using active drugs versus zero for placebo. Cure is defined as the reversion of cognition to normal levels in standard tests such as Consortium to Establish a Registry for Alzheimer’s Disease (CERAD) and the *Mini-Mental State Examination* (*MMSE*). The number in each group would be calculated on the basis of ≥30% more patients experiencing reversal of AD from active drugs than from placebo, and an annual drop-out rate of 10%. The planned duration of treatment would be three years with an annual review of data by a study board (DSMB). Participants would be aged ≥70, with a diagnosis of AD made according to standard criteria [177]. Exclusions would be individuals with particular risk factors for AD, viz., type-1 diabetes, untreated hypertension (BP ≥ 130/85), and untreated hepatitis C.

## 8. Conclusions and Summary

Curative treatment of AD should address the decreases in astrocytes, oligodendrocytes, neurons, and endothelial cells/pericytes, and increases in microglia, that occur in AD.

A two-drug combination might comprise pioglitazone with fluoxetine or pioglitazone with lithium; a three-drug combination might comprise those two drugs plus clemastine, fingolimod, or erythropoietin.

Low doses of those drugs will reduce SAEs and drug–drug interactions.

Clinical trial will demonstrate the validity of the proposed treatment.

## Data Availability

Not applicable.

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
