# Peer review of "Cure of Alzheimer’s Dementia Requires Addressing All of the Affected Brain Cell Types"

_jcm, 2023, doi:10.3390/jcm12052049_

Round 1
Reviewer 1 Report
The author presented a review article on the interesting topic of biological background for AD combination therapy. Unlike other review articles on combination therapy for AD, he proposed basing the choice for therapy on the types of brain cells involved.
In addition to neurons, the cells he estimated to be the most important are glial and endothelial cells. Examining the modifications and the possible biological role of these cells he proposed five therapeutic agents. The article could be read as a premise to motivate clinical trials with combinations of the suggested drugs.
It is indeed an interesting point of view backed by biological plausibility, albeit with little data from human studies. The author has published other interesting papers on the same topic, albeit with a different approach (one of these is rightly mentioned among the references of the article).
A superficial reading and interpretation of the article is to be avoided as the suggested drugs are already prescribed for various pathologies and consequently one could be induced to use them off labelI think there is no data to support an off-label use of the combination therapy reviewed here, and this should be highlighted in the article. For example in chapter 8 Conclusions and summary: "..This article does not intend to suggest the off label use of these therapies, but only the need for a clinical trial to demonstrate the validity of the proposed treatment" or something similar.
Some detailed comments :
64 – 65 …. who are APOε4 car-64 riers, have abnormally low rates of glucose metabolism in the neocortex that may also be 65 present in non-demented carriers of APOε4… : it is true that you discussed later on the importance of APOe4 but the specific issue is not discussed anymore . I suggest introducing a reference on it ( for example : Norwitz NG, Saif N, Ariza IE, Isaacson RS. Precision Nutrition for Alzheimer's Prevention in ApoE4 Carriers. Nutrients. 2021 Apr 19;13(4):1362. doi: 10.3390/nu13041362. PMID: 33921683; PMCID: PMC8073598. Or other of your choice)
132 …4. In summary, the affected brain cell-types in AD are astrocytes, oligodendrocytes, neurons, endothelial cells/pericytes, and microglia… I suggest adding APOe4 in the paragraph title
172 by excessive glutamate [35]Yuan/Wang [91]; something wrong with the reference
178 [105]; the article is not on the topic, check the reference
182 .. Fingolimod, an agonist of sphingomyelin phosphate : I find this definition insufficient, because “….The immunomodulator fingolimod is the prodrug of an S1P receptor agonist…” Brunkhorst R, Vutukuri R, Pfeilschifter W. Fingolimod for the treatment of neurological diseases-state of play and future perspectives. Front Cell Neurosci. 2014 Sep 12;8:283. doi: 10.3389/fncel.2014.00283. PMID: 25309325; PMCID: PMC4162362.
212 increased antioxidants[97,105],.. but in the reference article 105 I did not find specific mention of antioxidants increase activity.
228 … because memantine up-regulates the M2 isotype[127] …. wrong reference
332- 333 : …it is important to note that this use of the suggested drugs would be off-label, and that clinical 333 trial is required to establish their efficacy for reversal of AD….. I strongly suggest adding a sentence explaining that the article is not intended as a suggestion for off label use
Reviewer 2 Report
The manuscript is interesting, detailed, and would be of practical use, but in order to meet the criteria for publication, it requires serious technical correction. There are many technical errors in the manuscript that must be corrected (see the comments in the pdf document).

Reviewer 3 Report
The paper represents the original pathogenetic approach for the perspectives in treatment of AD based on huge literature review devoted to neuro-glial and neuro-vascular interconnections and interrelations. The author's choice of drug combinations probably effective for the AD treatment suggested available, but requires special clinical trials to be proved. The paper represents both fundamental and practical interest for specialists in the fields of biological and clinical psychiatry and psychopharmacology. The manuscript meets no objections and the article is recommended for publication in present form.
